# Variation in the modality of a yeast signaling pathway is mediated by a single regulator

**Julius Palme[1†], Jue Wang[2†], Michael Springer[1]\***

[1]Department of Systems Biology, Harvard Medical School, Boston, United States; [2]Department of Chemical Engineering, University of Washington, Seattle, United States

**Abstract** Bimodal gene expression by genetically identical cells is a pervasive feature of signaling networks and has been suggested to allow organisms to hedge their 'bets' in uncertain conditions. In the galactose-utilization (GAL) pathway of *Saccharomyces cerevisiae*, gene induction is unimodal or bimodal depending on natural genetic variation and pre-induction conditions. Here, we find that this variation in modality arises from regulation of two features of the pathway response: the fraction of cells that show induction and their level of expression. GAL3, the galactose sensor, controls the fraction of induced cells, and titrating its expression is sufficient to control modality; moreover, all the observed differences in modality between different pre-induction conditions and among natural isolates can be explained by changes in GAL3's regulation and activity. The ability to switch modality by tuning the activity of a single protein may allow rapid adaptation of bet hedging to maximize fitness in complex environments.

**\*For correspondence:**
michael_springer@hms.harvard.edu

[†]These authors contributed equally to this work

**Competing interests:** The authors declare that no competing interests exist.

## Introduction

Non-genetic heterogeneity is a pervasive feature of gene expression and cellular signaling (*Kærn et al., 2005*; *Balázsi et al., 2011*; *Raj and van Oudenaarden, 2008*). Bimodal responses, where cells in an isogenic population adopt one of two distinct states, are particularly important for microbes coping with fluctuating environments (*Grimbergen et al., 2015*; *Veening et al., 2008*) and cells of multicellular organisms differentiating into discrete types (*Xiong and Ferrell, 2003*; *MacArthur et al., 2009*).

The galactose-utilization (GAL) pathway in *Saccharomyces cerevisiae* is a well-characterized bimodal response and a classic model of microbial decision-making (*Johnston, 1987*; *Bhat, 2008*). Bimodality of GAL gene expression has been attributed to bistability arising from positive feedback through the Gal1p kinase and the Gal3p transducer (*Venturelli et al., 2012*; *Acar et al., 2005*). Perturbations of many of the components of the GAL pathway such as the Gal2p permease, the Gal4p activator, and the Gal80p repressor have been found to affect quantitative features of the GAL response (*Acar et al., 2005*; *Hawkins and Smolke, 2006*; *Acar et al., 2010*; *Ramsey et al., 2006*) and in principle could modify the feedback in the system and thus affect whether the response is bimodal or unimodal. However, only changes in Gal1p and Gal3p *Venturelli et al., 2012*; *Acar et al., 2005* have been shown to affect modality.

Our existing insight into modality in the GAL system comes almost entirely from measuring one pathway phenotype, the induced fraction, under one environmental perturbation, galactose titration (*Venturelli et al., 2012*; *Acar et al., 2010*; *Venturelli et al., 2015*; *Peng et al., 2015*; *Lee et al., 2017*). The few studies that have deviated from this experimental approach have resulted in observations that raise new questions. For example, the GAL response was found to be unimodal or bimodal depending on the carbon source prior to encountering galactose (*Biggar and Crabtree,*

*2001*); the molecular basis of this behavior is unknown. In our own previous work that focused on differences in GAL genes induction in mixtures of glucose and galactose between natural isolates (*Lee et al., 2017*; *Wang et al., 2015*), we noted that some strains showed a bimodal response while others had a unimodal response (*Lee et al., 2017*). These differences provide an opportunity to dissect the genetic basis underlying the differences in modality.

In this work, we confirm and expand the observation that the pattern of GAL pathway induction can be either unimodal or bimodal depending on genetic background and pre-induction conditions. A phenomenological model of GAL induction led us to a conceptual framework for variation in modality that identified the relative thresholds of induced fraction and expression level regulation as the two critical factors. Using this simple framework, we can explain the variation in modality we observed and predict how new perturbations would affect modality. Finally, we show that both natural variation and pre-induction conditions achieve changes in modality by tuning the expression and activity of a single signaling protein in the GAL pathway. These results reveal a simple evolutionary mechanism by which organisms can shape their responses to the environment, and suggest that modality is a highly adaptable feature of a signaling response.

## Results

### Genetic and environmental factors affect the modality of the GAL response

To study what causes the GAL response to be unimodal in some strain backgrounds and bimodal in other strain backgrounds, we measured the expression of a *GAL1* promoter driving YFP (GAL1pr-YFP) in 30 geographically and ecologically diverse yeast strains (*Wang et al., 2015*; *Liti et al., 2009*; *Cromie et al., 2013*) grown in different combinations of glucose and galactose (*Figure 1A*). We titrated glucose concentration in a constant concentration of galactose and observed that some displayed bimodal population distributions (*Figure 1B*, 'DBVPG1106') while others displayed unimodal population distributions with a glucose-dependent *GAL1* expression level (*Figure 1B*, 'BC187', *Figure 1—figure supplement 1*; *Ricci-Tam et al., 2021*). Both unimodal and bimodal population distributions were stable even when cells were kept in the same environment for 24 hr (*Figure 1—figure supplement 2*), suggesting that the difference in modality is a steady-state phenomenon. In addition, we studied what causes the modality of the GAL response to change based on growth history. *Biggar and Crabtree, 2001*, previously showed that populations of the laboratory strain S288C had a unimodal GAL induction when grown with raffinose as a carbon source prior to encountering mixtures of glucose and galactose, but a bimodal response when mannose was used as the initial carbon source (*Figure 1C*). In contrast to our results when titrating galactose in the presence of glucose, all 30 of our natural isolates showed bimodal responses when we titrated the galactose concentration in the absence of glucose (*Figure 1D*, *Figure 1—figure supplement 3*). These observations suggest that glucose plays a critical role as a second input to the bistable GAL pathway that can cause a qualitative change in the modality of GAL pathway induction.

### Differences in induction and expression level regulation can explain the variation of modality

In the absence of galactose, Gal80p binds the transcription factor Gal4p and keeps it in an inactive state (*Lue et al., 1987*; *Wu et al., 1996*). In the presence of galactose, Gal3p binds Gal80p, releasing Gal4p from the Gal80p-Gal4p complex, and allowing Gal4p to drive the transcription of a number of GAL genes. The initiation of transcription of the galactose sensors Gal1p and Gal3p by Gal4p creates a positive feedback loop, and this is believed to be the mechanism underlying the bistability of the system (*Venturelli et al., 2012*).

While glucose can completely inhibit the GAL response, its role in determining the modality of the response has been poorly explored. Glucose has two inhibitory effects on the GAL pathway. (1) It indirectly decreases the intracellular concentration of galactose through competition for binding to transporters (*Escalante-Chong et al., 2015*). This in turn decreases the amount of active Gal3p and thereby affects the fraction of cells that induce the GAL pathway (induced fraction) (*Ricci-Tam et al., 2021*). (2) It directly increases the activity of the transcriptional repressor Mig1p which regulates *GAL4* expression and thereby decreases the expression of GAL genes (induction level)

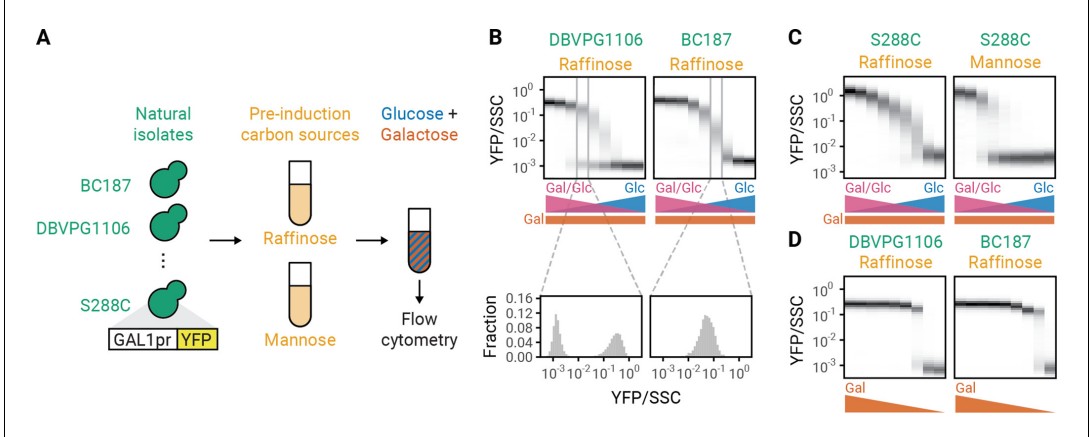

**Figure 1.** Genetic and environmental factors change galactose-utilization (GAL) modality. (**A**) Experimental workflow. Natural isolates of yeast tagged with a fluorescent reporter of *GAL1* (GAL1pr-YFP) expression were first grown in synthetic (S) medium with a pre-induction carbon source for 16 hr, then switched to S medium with mixtures of glucose and galactose. After 8 hr, *GAL1* expression was analyzed by flow cytometry. (**B–C**) GAL induction of two natural isolates (DBVPG1106 and BC187) or a lab strain (S288C) in mixtures of glucose and galactose after pre-induction growth in raffinose or mannose. Glucose concentration was titrated in twofold steps from 0.0039% to 1% while galactose concentration was kept constant at 0.25%. Top: Induction profiles of two natural isolates. Each plot is composed of nine histograms with color intensities corresponding to the density of cells with a given YFP abundance. Galactose concentration was titrated in twofold steps from 1% to 0.0039%. Bottom: Blow out of two histograms of YFP level normalized by side scatter (SSC) at a single concentration of glucose and galactose (see *Figure 1—figure supplement 4* for histograms of all conditions). Since the GAL pathway responds to the ratio of galactose and glucose (*Escalante-Chong et al., 2015*), increasing the glucose concentration while keeping the galactose concentration constant simultaneously decreases GAL activation and increases glucose repression. All measurements are representative examples of at least two independent repeats. Repeat measurements are plotted in *Figure 1—figure supplement 5*. (**D**) GAL induction of two natural isolates in different galactose concentrations after pre-induction growth in raffinose.

The online version of this article includes the following figure supplement(s) for figure 1:

**Figure supplement 1.** Galactose-utilization (GAL) induction of natural isolates in different glucose and galactose concentrations.

**Figure supplement 2.** Galactose-utilization (GAL) induction profiles of 14 natural isolates (panel titles) with unimodal induction behavior after 8 or 24 hr in the same environment.

**Figure supplement 3.** Galactose-utilization (GAL) induction of natural isolates in different galactose concentrations.

**Figure supplement 4.** Histograms of the glucose gradient heatmaps plotted in *Figure 1B and C*.

**Figure supplement 5.** Reproducibility of induced fraction (blue) and induced mean (green) measurements of different 30 natural isolates (panel titles).

(*Ricci-Tam et al., 2021*; *Figure 2A*). Because the system is bimodal in pure galactose, glucose cannot drive unimodality solely through the transporter-dependent indirect effect, leading us to suspect that the Mig1p-dependent mechanism is responsible.

We built a phenomenological model of the GAL pathway to determine whether independently tuning the indirect and direct effects of glucose are sufficient to change modality. Based on our measurements of the pathway response (*Ricci-Tam et al., 2021*), we mathematically described the indirect and direct effects of glucose as Hill functions that decrease with increasing glucose concentration, with the final induction profiles being a simple product of these composite functions. To simulate a range of population induction profiles, we generated population distributions for the induced fraction and induced level from a normal distribution whose standard deviation is derived from GAL gene expression measurements (*Figure 2—figure supplement 1*; Materials and methods). We then varied the glucose threshold for the induced level while keeping the glucose threshold for the induced fraction constant (*Figure 2B*) or the glucose threshold for the induced fraction while keeping the glucose threshold for the expression level constant (*Figure 2C*). Indeed, changing either can switch the population behavior between unimodal and bimodal. In both cases, the pathway is bimodal when the glucose inhibition threshold for the induction level is less than the glucose inhibition threshold for the induced fraction.

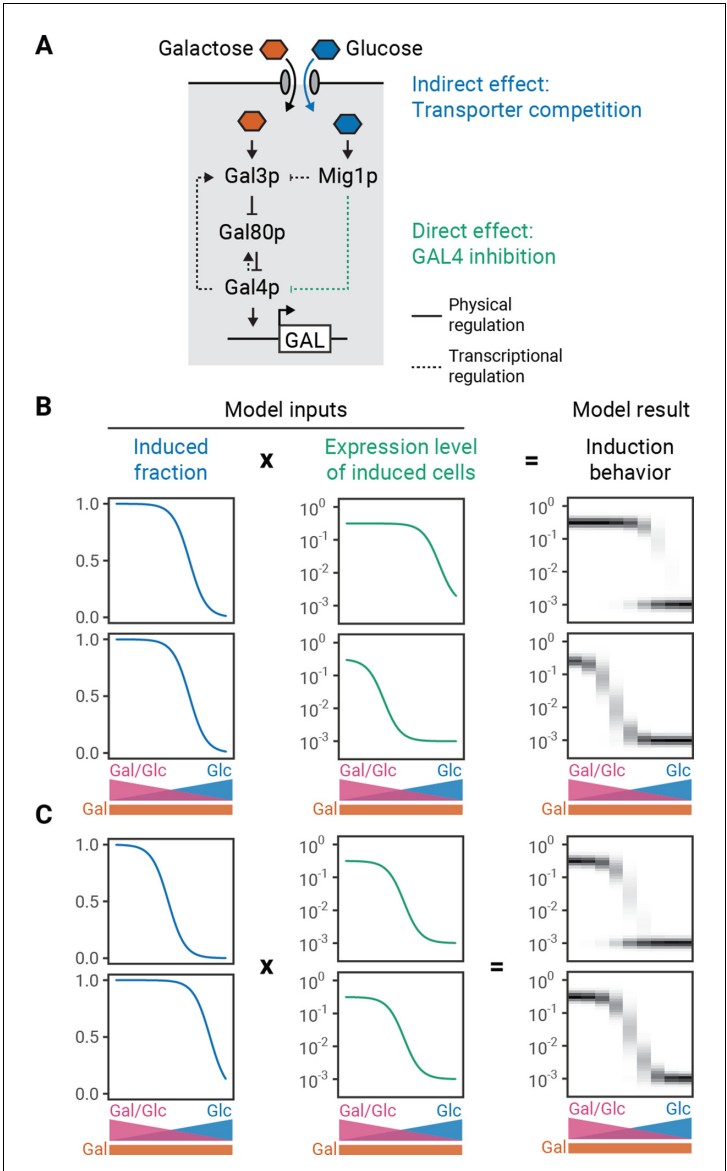

**Figure 2.** Phenomenological modeling of galactose-utilization (GAL) induction. (**A**) Schematic description of the GAL pathway with indirect or direct inhibition by glucose. (**B–C**) Modeling results for varying expression level regulation with constant induced fraction regulation (**B**) or varying induced fraction regulation with constant expression level regulation (**C**). The induced fraction and mean induced level curves were chosen to be Hill functions based on empirical data (e.g. *Figure 1—figure supplement 5*). To create a population distribution, normal distributions were defined around the mean (log) induced level and a constant uninduced expression level, with standard deviations determined by fitting to observed distributions (*Figure 2—figure supplement 1*). The overall expression distribution is the induced-fraction-weighted sum of induced and uninduced distributions. For the uninduced subpopulation, the relative expression level was set to $10^{-3}$. Model results are represented by histograms at nine different glucose and galactose combinations. The intensity of the color on the plot corresponds to the density of cells with a given induction value. This analysis can be extended to a continuous range of induced fraction and induced level behaviors (*Figure 2—figure supplement 2*).

The online version of this article includes the following figure supplement(s) for figure 2:

**Figure supplement 1.** Fit of the standard deviation to the mean induction level.

**Figure supplement 2.** Comparison of a wide range of induced fraction and induced level behaviors.

## Differences in induction and expression level regulation predict the modality of natural isolates

To analyze whether natural genetic variation could change modality by altering the induced fraction or induction level, we analyzed these features in our 30 natural yeast isolates. First, we determined the induced and uninduced subpopulations by comparing GAL reporter distributions to the distribution of an uninduced sample (as in *Peng et al., 2015*, *Figure 3—figure supplement 1*). From these two subpopulations, we then calculated two summary metrics for each strain's behavior: $E_{10}$ ('expression level threshold'), the glucose concentration where the GAL1 expression level of the induced subpopulation reaches 10% of its level in pure galactose (*Figure 3A*), and $F_{90}$ ('induced fraction threshold'), the glucose concentration where 90% of cells are in the induced subpopulation (*Figure 3B*). Our modeling suggests that determining the $E_{10}$ and $F_{90}$ should be sufficient to predict whether the induction behavior is bimodal or unimodal. To test this hypothesis, we used the measured $E_{10}$ and $F_{90}$ as inputs into phenomenological model and computed modality. Overall, for both bimodal and unimodal strains, the experiments and models are in good agreement (*Figure 3C and D*, *Figure 3—figure supplements 2–3*). One minor discrepancy is that metrics from unimodal strains often predict a narrow range of bimodality in our model. We believe there are two factors that might explain the difference between the experimental data and the model predictions. First, to measure the $F_{90}$, there must be measurable gene induction. Therefore, the calculated $F_{90}$ is a lower bound for the actual value of $F_{90}$ in unimodal strains; in many cases, the calculated $F_{90}$ will be higher than the actual $F_{90}$. If the calculated $F_{90}$ for simulations of unimodal strains is increased by even a factor of 2, the discrepancy between the behaviors disappears (*Figure 3—figure supplement 4*). Second, the slopes of the induced fraction curves could also be subject to variation and this could affect modality. Increasing the steepness of the induced fraction curve or induced level curves by increasing the Hill coefficients in our models can make simulations more unimodal (*Figure 3—figure supplement 5*). Indeed, some natural isolates appear to have steeper induction curves (*Figure 3—figure supplement 6*). Overall, however, our model correctly predicts and provides a useful conceptual lens for understanding bimodality of GAL pathway induction across natural isolates.

We next scanned the parameters for our phenomenological model using a wide range of summary metrics and slopes to delimit a phase diagram of GAL induction modality. We then compared this phase diagram to the experimental data from natural isolates (*Figure 3E*). The modality of the natural isolates agrees well with their predicted modality in the phase diagram, supporting our hypothesis that the $E_{10}$ and $F_{90}$ measurements capture the important biological features that determine the modality of a strain.

Our phenomenological model gives us a potential molecular explanation for unimodality: Mig1p activity leads to unimodality by inhibiting *GAL4* expression in the regime where Gal3p activation is still strongly dependent on glucose and galactose. Therefore, a strong prediction of the model is that removing Mig1p regulation should restore bimodality in unimodal strains. To test this prediction, we analyzed the induction profile of a *mig1Δ* strain. Deleting *MIG1* removes the glucose dependent regulation of *GAL4* expression level and thus the $E_{10}$ of the deletion strain is increased compared to the wild-type strain (*Figure 3F*). As predicted, deleting *MIG1* converts the strain from unimodal to bimodal. In addition, the observed $F_{90}$ value for the bimodal *mig1Δ* strain is higher than that for the unimodal wild type. This supports our hypothesis that Mig1p-dependent repression conceals the actual $F_{90}$ value of unimodal strains and that the observed $F_{90}$ value of unimodal strains is a lower bound for the actual $F_{90}$ value.

## Differences in induction regulation explain the history dependence in modality

It has previously been reported that pre-induction growth conditions can affect the modality of GAL induction (*Biggar and Crabtree, 2001*). This offers another opportunity to test the predictions from our modeling framework. To see how metabolic history affects $F_{90}$ and $E_{10}$, we grew 13 natural isolate strains in mannose, raffinose, acetate, or glycerol prior to transferring them into mixtures of glucose and galactose (*Figure 4—figure supplement 1*). A range of different behaviors were observed, which could be broken into two categories of responses. The first category is strains that are unimodal in some pre-induction conditions but bimodal in others (*Figure 4A and C*). The second category is strains that do not change modality based on the tested pre-induction conditions; the strains are

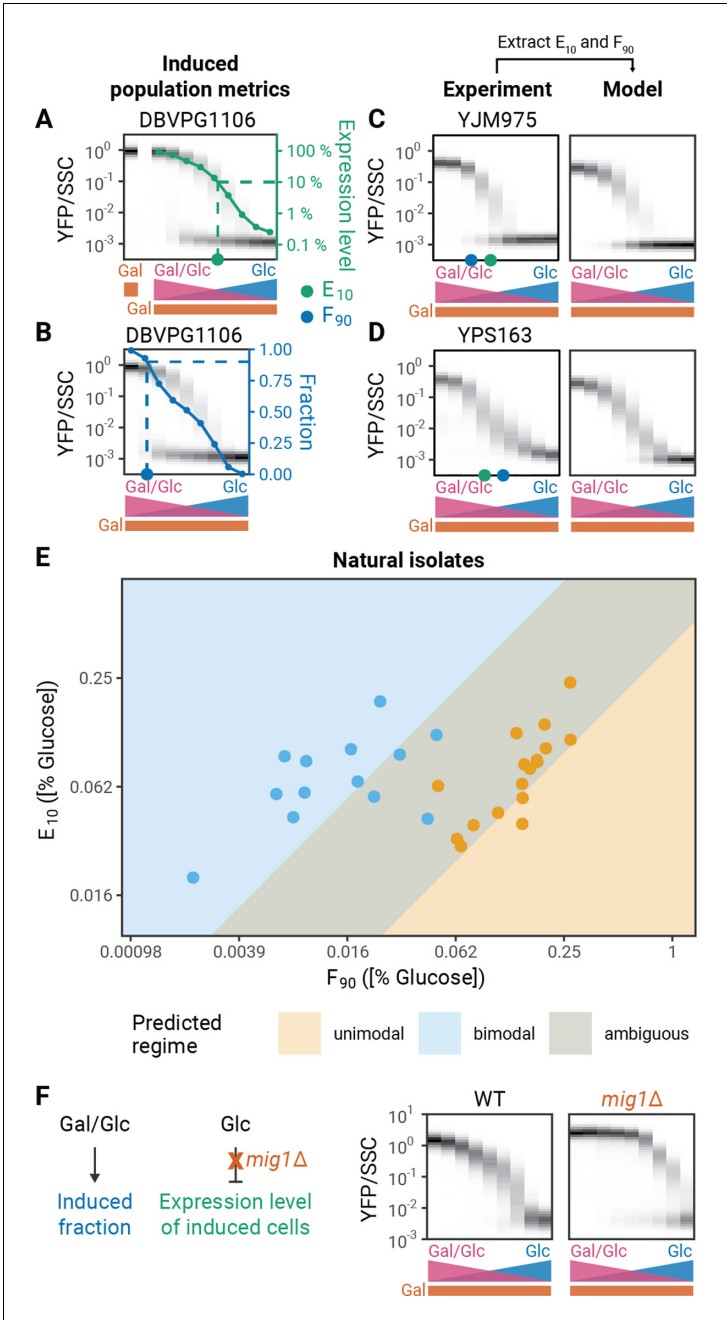

**Figure 3.** Experimental validation of model predictions. (**A**) Expression level metric ($E_{10}$). GAL induction is measured in 2% galactose to determine the maximal expression level. The $E_{10}$ is the glucose concentration at which the expression level of the induced subpopulation reaches 10% of the expression level in 2% galactose in the absence of glucose. (**B**) Induced fraction metric ($F_{90}$). The induced fraction for each strain is calculated as the fraction of cells with an expression level that is outside the range of an uninduced population grown in 2% glucose (*Figure 3—figure supplement 1*). The $F_{90}$ is the glucose concentration where the induced fraction reaches 90%. (**C–D**) Modality prediction based on induction metrics. In the phenomenological model, the position of the induced fraction and induced level functions are determined by the $F_{90}$ and the $E_{10}$, respectively. (**E**) $F_{90}$ and $E_{10}$ values of a panel of 30 natural isolates. Values correspond to the mean of two to five replicates. Modality was determined by comparing the fit of the data to a single or double Gaussian model (See Materials and methods). Background colors correspond to predicted regimes for unimodal and bimodal strains. Simulations with diverse combinations of $F_{90}$ and $E_{10}$ values as well as different slopes for the induced fraction and induced level curves delineate regimes of unimodal and bimodal behaviors. The overlap represents an ambiguous regime where both

*Figure 3 continued on next page*

*Figure 3 continued*

unimodal and bimodal behaviors are possible. Standard deviations of the $F_{90}$ and $E_{10}$ measurements are plotted in *Figure 3—figure supplement 7*. (F) Effect of *mig1Δ* on *GAL1* induction profiles.

The online version of this article includes the following figure supplement(s) for figure 3:

**Figure supplement 1.** Identification of the induced subpopulation.

**Figure supplement 2.** Experimental and simulated galactose-utilization (GAL) induction profiles of bimodal strains.

**Figure supplement 3.** Experimental and simulated galactose-utilization (GAL) induction profiles of unimodal strains.

**Figure supplement 4.** Experimental and simulated galactose-utilization (GAL) induction profiles of unimodal strains with higher $F_{90}$.

**Figure supplement 5.** Experimental and simulated galactose-utilization (GAL) induction profiles of unimodal strains with varying steepness.

**Figure supplement 6.** Fitted values for the n parameter.

**Figure supplement 7.** Effect of the threshold for the size of the smaller subpopulation on the modality metric.

**Figure supplement 8.** $F_{90}$ and $E_{10}$ values of a panel of the 30 natural isolates in *Figure 3*.

always bimodal or always unimodal in all pre-induction carbon sources tested here (*Figure 4E*). As predicted by the model, pre-induction conditions led to changes in $F_{90}$ and/or $E_{10}$ that should change modality; all our experimental results agree with the predictions from our phase diagram of GAL induction (*Figure 4B,D and F*). While changing either $F_{90}$ or $E_{10}$ in model is sufficient to change modality, we found that in response to changes in pre-induction carbon the change in $F_{90}$ was considerably larger than the change in $E_{10}$ (8.6 versus 1.8, respectively; *Figure 4—figure supplement 2*). Our phenomenological model predicts that the magnitude of the changes in $F_{90}$ alone are sufficient to explain the observed changes of modality.

## Differences in *GAL3* expression explain the history dependence in modality

To determine how pre-induction carbon source modulates $F_{90}$, we measured the expression of GAL genes in pre-induction conditions using transcriptional reporters (*Figure 5A*). We found that GAL genes are down-regulated in carbon sources that lead to bimodal induction (mannose) and up-regulated in carbon sources that lead to unimodal induction (acetate, glycerol). Among all GAL genes, the expression levels of *GAL3* and *GAL4* show the strongest fold change between the carbon sources tested (*Figure 5A*).

We hypothesized that *GAL3* was more likely than *GAL4* to be the dominant factor due to its high dynamic range of expression (*Figure 5A*) and prior evidence that *GAL3* can have a large effect on the GAL decision (*Acar et al., 2010*). We therefore analyzed the regulation of *GAL3* by a range of pre-induction carbon sources in nine natural isolates using a transcriptional reporter for the *GAL3*[S288C] promoter. We found that in each strain the *GAL3* expression level in a pre-induction conditions generally correlates with the modality observed later (*Figure 5B*).

To test if *GAL3* expression prior to induction is the key determinant of modality, we used a tetracycline-inducible promoter to control the expression of *GAL3* directly. We predicted that forcing a change in *GAL3* expression while keeping the pre-induction carbon the same should change modality. Conversely, changing the pre-induction carbon without changing *GAL3* expression should not change modality.

Indeed, we found that the pre-induction level of Gal3p, not the pre-induction carbon, is critical for setting modality. For the laboratory strain S288C, pre-induction growth in mannose leads to low *GAL3* expression and a bimodal induction profile, while pre-induction growth in raffinose leads to higher *GAL3* expression and a unimodal induction profile (*Figures 4A* and *5A*). When we overexpressed *GAL3* during pre-induction growth in mannose using tetracycline induction, we saw an increase in the induced fraction and a loss of bimodality (*Figure 5C*). Thus, the *GAL3* concentration pre-induction is sufficient to set the modality in this strain background. Similarly, artificially setting the *GAL3* level of mannose pre-induction cultures to that of a raffinose pre-induction culture converted the induction profiles to one similar to a raffinose pre-induction culture (*Figure 5C,II* and *IV*). In addition to showing unimodal induction after raffinose pre-induction, this strain also shows

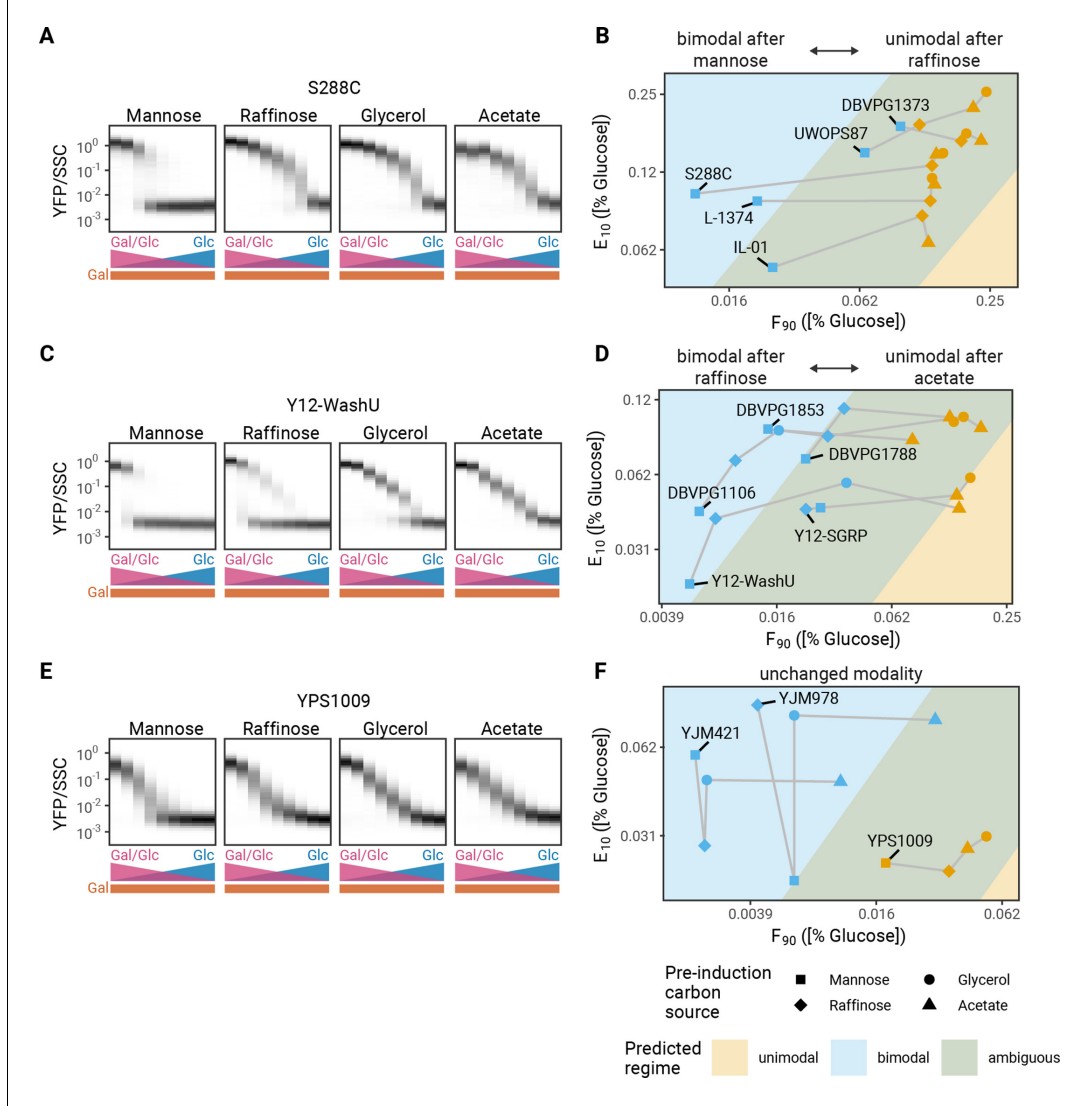

**Figure 4.** Metabolic history changes modality. (**A**) History dependence of induction profiles. Induction profiles of S288C in mixtures of glucose and galactose after pre-induction growth in different carbon sources for 16 hr. (**B**) $F_{90}$ and $E_{10}$ values for isolates that are unimodal after growth in raffinose and bimodal after growth in mannose. (**C**) Induction profiles of Y12-WashU after growth in different carbon sources for 16 hr. (**D**) $F_{90}$ and $E_{10}$ values for isolates that are bimodal after growth in raffinose and unimodal after growth in either acetate or glycerol. (**E**) Induction profiles of YPS1009 after growth in different carbon sources for 16 hr. (**F**) $F_{90}$ and $E_{10}$ values for isolates that are always unimodal or bimodal. All measurements are representative examples of two independent repeats (compared in *Figure 4—figure supplements 3–4*).

The online version of this article includes the following figure supplement(s) for figure 4:

**Figure supplement 1.** Galactose-utilization (GAL) induction of natural isolates in different glucose and galactose concentrations after growth in different carbon sources.

**Figure supplement 2.** Fold change between the highest and lowest $E_{10}$ and $F_{90}$ values after growth in different pre-induction conditions for all isolates shown in *Figure 4—figure supplement 1*.

**Figure supplement 3.** Reproducibility of induced fraction (blue) and induced mean (green) measurements of 14 natural isolates (left titles) after growth in different pre-induction carbon sources (top titles).

**Figure supplement 4.** Reproducibility of $F_{90}$ and $E_{10}$ measurements.

unimodal behavior after pre-induction in acetate or glycerol (*Figure 4A*). Next, we titrated *GAL3* in a strain where the endogenous *GAL3* gene was deleted to allow for constant *GAL3* expression below the wild-type level. As predicted, when pre-induction *GAL3* expression in these sugars was low, we saw bimodal induction almost identical to that seen with mannose pre-induction (*Figure 5D*). Pre-induction *GAL3* concentrations also set the induced fraction with almost no

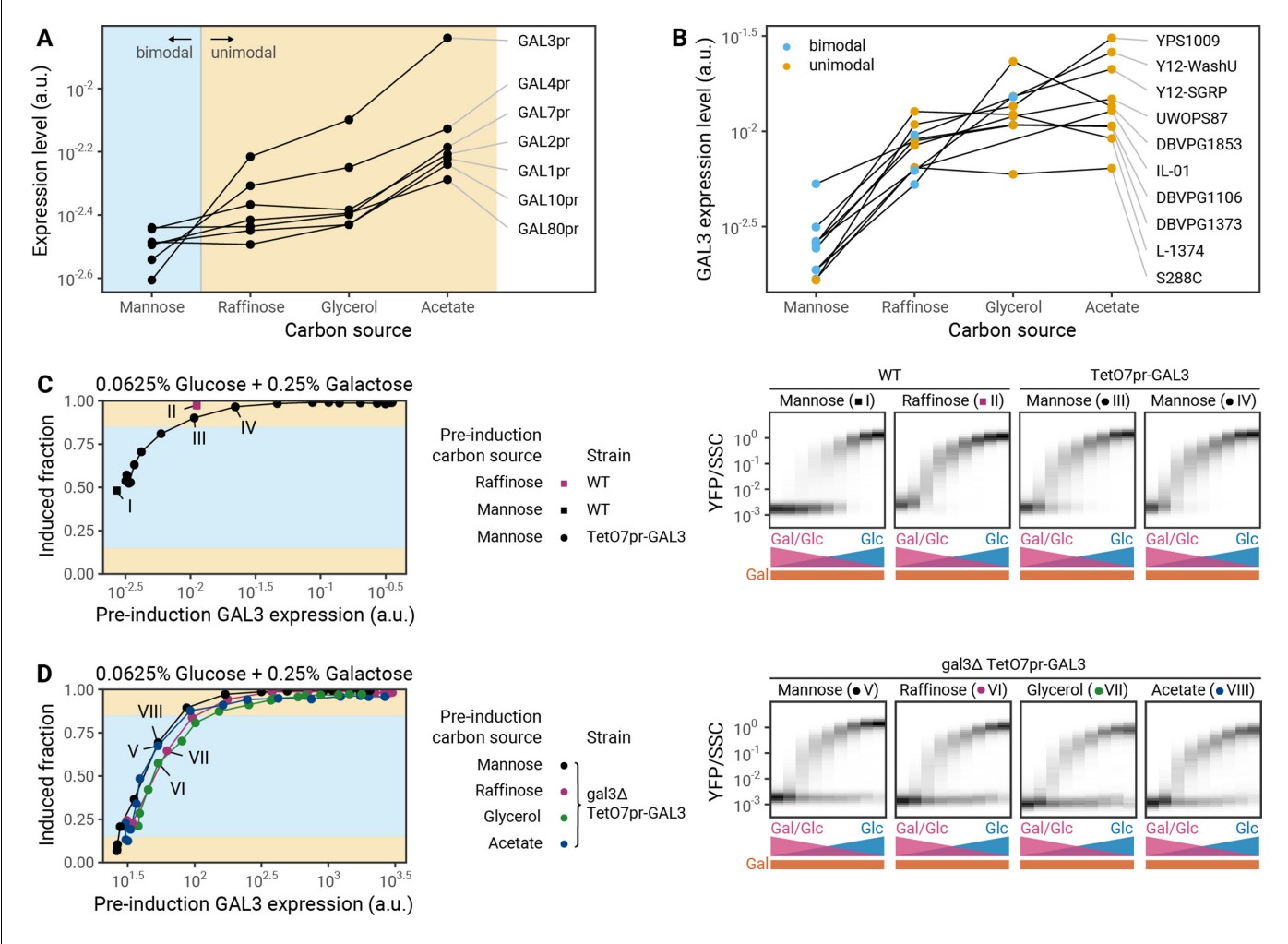

**Figure 5.** Pre-induction *GAL3* levels determine the modality of induction profiles. (A) Expression of galactose-utilization (GAL) genes in different pre-induction carbon sources in S288C as determined by fluorescence of GAL promoter-YFP protein transcriptional reporter strains. (B) Effect of pre-induction growth in different carbon sources on the expression of a *GAL3*^S288C reporter in S288C and nine natural isolates. Colors correspond to the modality of the induction profile of these strains in the given pre-induction carbon source. (C) Left: Effect of *GAL3* overexpression during pre-induction growth in mannose. Connected points correspond to a doxycycline titration series for a S288C TetO7pr-*GAL3* strain. Right: Complete induction profiles of S288C in mannose (I) or raffinose (II) and S288C TetO7pr-*GAL3* in mannose at two concentrations of doxycycline that lead to *GAL3* expression levels that bracket the expression level of GAL3 from a raffinose pre-induction culture (III, IV). (D) Left: Effect of synthetic *GAL3* expression in a Δ*gal3* during pre-induction growth in different carbon sources. Connected points correspond to a doxycycline titration series in different carbon sources for a S288C Δ*gal3* TetO7pr-GAL3-mScarlet strain. Right: Complete induction profiles of S288C Δ*gal3* TetO7pr-GAL3-mScarlet after pre-induction growth in different carbon sources with constant *GAL3* expression.

dependence on the pre-induction carbon source (*Figure 5D*). We conclude that regulation of *GAL3* expression in pre-induction conditions is the major driver of history dependence in the modality of GAL induction.

## Natural variation in *GAL3* alleles underlies the genetic changes in modality

The central role of *GAL3* expression in setting the modality of induction suggests that natural variation in *GAL3* alleles could be responsible for the observed differences in modality between isolates (*Figure 1B*). Previously, we showed that polymorphisms in the *GAL3* gene explain most of the natural variation in the decision to induce the GAL pathway (i.e. the $F_{90}$) (*Lee et al., 2017*), suggesting that allele swaps of the *GAL3* ORF should alter the $F_{90}$ of the strain, which in some cases would be

enough to switch the modality of induction. To test this prediction, we determined the modality of a set of 30 allele swap strains comprised of 10 *GAL3* alleles in three genetic backgrounds. The experimentally determined modality of all allele swap strains agrees with the expected modality

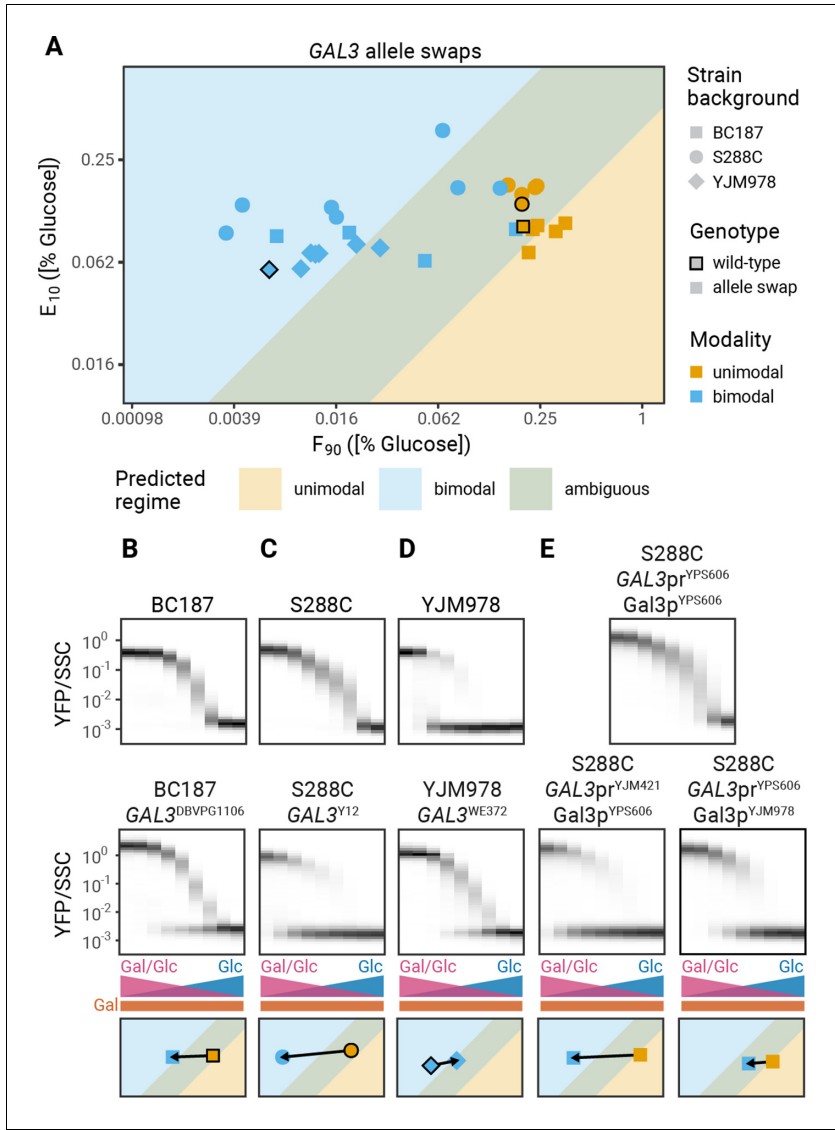

**Figure 6.** Allele swaps of *GAL3* change modality. (A) $F_{90}$ and $E_{10}$ values for a panel of allele swaps (10 *GAL3* alleles in three different genetic backgrounds). Strains that never reach an induced fraction of 90% are not shown here. Black outlines denote the wild-type strains. All measurements are representative examples of two independent repeats (compared in *Figure 6—figure supplements 4–5*). (B–E) Effect of *GAL3* allele swaps on modality. (Top) Induction profiles of (B–D) wild-type isolates or (E) S288C *GAL3*pr$^{YPS606}$. (Middle) Induction profiles of (B–D) *GAL3* allele swaps and (E) *GAL3* promoter or *GAL3* coding regions (CDS) swaps. (Bottom) Effect of the perturbation on $F_{90}$ and $E_{10}$. Arrows start at the values of the wild-type or S288C *GAL3*$^{YPS606}$ strains and end at the values of the perturbed strain.

The online version of this article includes the following figure supplement(s) for figure 6:

**Figure supplement 1.** Effect of *GAL1* and *GAL4* allele swaps.

**Figure supplement 2.** Effect of *GAL1* and *GAL4* allele swaps in the *GAL3* allele swap background.

**Figure supplement 3.** Reproducibility of induced fraction (blue) and induced mean (green) measurements of 10 *GAL3* alleles (left titles) in three different strain backgrounds (top titles).

**Figure supplement 4.** Galactose-utilization (GAL) induction of allele swap strains in different glucose and galactose concentrations.

**Figure supplement 5.** Reproducibility of $F_{90}$ and $E_{10}$ measurements.

(*Figure 6A*, *Figure 6—figure supplement 1*). For example, replacing the *GAL3* allele of two unimodal strains, BC187 and S288C, with the *GAL3* allele of any of the bimodal strains is sufficient to change the induction profiles from unimodal to bimodal (*Figure 6B and C*). In agreement with our model, the $F_{90}$ in these strains decreases sufficiently to convert strains from unimodal to bimodal. Replacing the *GAL3* gene of the bimodal strain YJM978 with the alleles of unimodal strains does not change the modality (*Figure 6D*). This is also in agreement with the model which predicts that the magnitude of the increase in $F_{90}$ caused by allele swaps in the YJM978 strain background is insufficient to change modality (*Figure 6D*). Swapping the alleles of *GAL1*, the second galactose sensor, or *GAL4*, the transcription factor that activates GAL gene expression, does not affect modality (*Figure 6—figure supplement 2*). Similarly, swaps of *GAL1* or *GAL4* alleles in *GAL3* allele swap strains have no additional effect on modality (*Figure 6—figure supplement 3*), suggesting that the role of *GAL3* in setting modality can evolve independently of other *GAL* genes.

To further explore the variations in modality among the natural isolates, we analyzed the contribution of promoter and coding sequence variation. We found that SNPs (single nucleotide polymorphisms) in either promoter or coding regions (CDS) are sufficient to change modality (*Figure 6E–F*). For example, S288C with *GAL3*$^{YPS606}$ is unimodal, but replacing the YPS606 promoter in this strain with the YJM421 promoter leads to bimodal induction (*Figure 6E*). Similarly, BC187 with *GAL3*$^{NC-02}$ is unimodal, but replacing the NC-02 coding sequence with the YJM978 coding sequence leads to bimodal induction (*Figure 6F*). The mechanisms by which promoter and CDS changes are able to change the modality will be the subject of future work.

## Discussion

In mixtures of glucose and galactose, the response of the GAL pathway in a population of yeast cells can be either unimodal or bimodal depending on their evolutionary history and their current environmental conditions. Here, we show that the modality of GAL induction in different strains depends on the relationship between the glucose effects on the induced fraction and the expression level. Glucose inhibits the expression of GAL genes, preventing the positive feedback that is crucial for bistability. In general, any input to a pathway that can conditionally create or eliminate feedback has the potential to modulate bistability. Since many signaling responses are controlled by multiple inputs, our findings imply that other unimodal responses could be bimodal in different conditions and vice versa.

Bimodality in the GAL response is considered a bet-hedging strategy where a fraction of the population prepares for glucose depletion by inducing the GAL pathway while other cells maximize their current growth and do not induce the pathway (*Venturelli et al., 2015*; *Wang et al., 2015*). This heterogeneity helps populations deal with uncertain, fluctuating environments. Bet-hedging is advantageous in the GAL system when the switching rate between glucose and galactose environments is high (*Acar et al., 2008*). Indeed, cells evolve bimodality in MAL gene expression when they are continuously switched between glucose and maltose (*New et al., 2014*). Because cells can sometimes be in environments with a high switching rate and other times in environments with a low switching rate, a strategy that allows the extent of bet-hedging to be tuned could be optimal. In this work, we show strains can tune the amount of bimodality both physiologically, based on their metabolic history, and genetically, presumably based on the environmental statistics that different natural isolates have faced in their evolutionary history. Further work will be needed to determine the evolutionary consequences of tunable bimodal responses such as the ones we characterize here.

Previous work on cell-to-cell heterogeneity has typically emphasized the complex genetic architecture of the pathways involved (*Ansel et al., 2008*; *Fehrmann et al., 2013*). In contrast, the physiological and genetic variation in modality in the GAL pathway can be explained by changes in the behavior of a single gene. Swapping the *GAL3* alleles of natural isolates can turn a unimodal strain into a bimodal strain. We show that the environment tunes the expression level of *GAL3*, and this tuning is sufficient to change the modality of GAL induction (*Figure 6* and *Figure 6—figure supplement 1*). Circuit designs such as these, where a single gene controls modality, may have been selected in evolution, since they allow cells to easily adapt their behavior on both physiological and evolutionary timescales.

The control of GAL pathway modality by mannose, raffinose, glycerol, and acetate suggests an additional layer of metabolic regulation that has been largely missed in previous analyses of this

pathway. These findings show that factors other than canonical glucose catabolite repression can be important in determining the inducibility of GAL genes, consistent with our findings that many mutants outside the GAL pathway can have a significant effect on GAL response (*Hua and Springer, 2018*). The fact that pre-induction carbon sources mostly affect $F_{90}$, just as *GAL3* allele swaps do, suggests that the *GAL3* positive feedback loop may be a nexus of regulation of GAL genes by multiple signals in the cell. In future studies, understanding the metabolic regulation of this well-studied system could give insight into the connections between metabolism and metabolic signaling in a variety of systems.

## Materials and methods

### Strains and media

Strains were obtained as described in *Lee et al., 2017*; *Wang et al., 2015*. An initial set of 36 strains were assayed in a glucose gradient (1–0.0039%) with a constant background of 0.25% galactose. Strains DBVPG6765, CLIB324, L-1528, M22, W303, YIIC17-E5 were excluded from downstream analysis due to poor growth in our media conditions. Strain 378604X was also excluded due to a high basal expression phenotype that was an outlier in our collection. The genetic basis of this behavior is likely an interesting topic for follow-up studies. All experiments were performed in synthetic minimal medium ('S'), which contains 1.7 g/L yeast nitrogen base (YNB) (BD, Franklin Lakes, NJ) and 5 g/L ammonium sulfate (EMD). In addition, D-glucose (EMD, Darmstadt, Germany), D-galactose (MilliporeSigma, St. Louis, MO), mannose (MilliporeSigma), glycerol (EMD), acetate (MilliporeSigma), and/or raffinose (MilliporeSigma) were added as a carbon source. Cultures were grown in a humidified incubator (Infors Multitron, Bottmingen, CH) at 30°C with rotary shaking 999 rpm (500 µL cultures in 1 mL 96-well plates).

### Flow cytometry

Cells were struck onto YPD agar from −80°C glycerol stocks, grown to colonies, then inoculated from colony into YPD liquid and cultured for 16–24 hr. Then, cultures were inoculated in a dilution series (1:200 to 1:6400) in S + 2% pre-induction carbon source medium. The pre-induction cultures were incubated for 16–20 hr, and then their optical density (OD600) was measured on a plate reader (PerkinElmer Envision). The outgrowth culture with OD600 closest to 0.1 was selected for each strain, and then washed twice in S (with no carbon sources). To determine expression levels in pre-induction conditions, washed cells were then diluted in Tris-EDTA pH 8.0 (TE) in a shallow microtiter plate (CELLTREAT, Pepperell, MA). For sugar gradient experiments, washed cells were diluted 1:200 into the appropriate sugar in 96-well plates (500 µL cultures in each well) and incubated for 8 hr. Then, cells were harvested and fixed by washing twice in TE and resuspended in TE before transferring to microtiter plate for measurement. Flow cytometry was performed using a Stratedigm S1000EX with A700 automated plate handling system.

### *GAL3* titration in pre-induction conditions

To titrate *GAL3* levels in the presence of the native *GAL3* gene, the *AGA1* gene was replaced with a MYO2pr-rtTA-TetO7pr-GAL3 construct in a hoΔ:GAL1pr-YFP strain. Cells were grown for 16 hr in S + 2% mannose as described above, but the medium was supplemented with doxycycline (MilliporeSigma) concentrations ranging from 38.9 to 0.0176 µg/mL in 1.5× dilutions steps. To measure the total *GAL3* expression level after pre-induction growth, the *AGA1* gene was replaced with a MYO2pr-rtTA-TetO7pr-YFP construct in a hoΔ:GAL3pr-YFP reporter strain. After pre-induction growth in the same dilution doxycycline concentrations, cells were harvested and YFP levels were determined using flow cytometry as described above.

To titrate *GAL3* levels in the absence of the native *GAL3* gene, the *AGA1* gene was replaced with a MYO2pr-rtTA-TetO7pr-GAL3-mScarlet construct in a gal3Δ hoΔ:GAL1pr-YFP strain. Cells were grown for 16 hr in S + 2% pre-induction carbon source as described above, but the medium was supplemented with doxycycline concentrations ranging from 38.9 to 0.0176 µg/mL in 1.5× dilutions steps. To measure the total *GAL3* expression level after pre-induction growth, cells were washed and mScarlet levels were determined by fluorescence microscopy using a Hamamatsu Orca-R2

camera (Hamamatsu, Japan) on a Ti Eclipse inverted Nikon microscope (Tokyo, Japan). Microscopy images were analyzed using U-net (*Falk et al., 2019*) and custom Python scripts.

## Data analysis

Data analysis was performed using custom MATLAB scripts, including Flow-Cytometry-Toolkit (https://github.com/springerlab/Flow-Cytometry-Toolkit); *Springer, 2016*.

To determine the modality of GAL induction experiments, a Gaussian function was fitted to the population distribution for each of the nine sugar combinations. If the degree-of-freedom adjusted $R^2$ of the fit was less than 0.99, two Gaussian functions were fitted to the data. Distributions were then determined to be bimodal if the distance between the means of the Gaussians was more than twice of the highest standard deviation of the Gaussian (as in *Venturelli et al., 2012*) and the fraction of the smaller Gaussian was higher than 0.15. The modality of induction profiles was mostly unaffected by changes in this threshold (*Figure 3—figure supplement 7*). GAL induction experiments or simulations that had a bimodal distribution in at least one combination of glucose and galactose in all replicates were called bimodal.

## Phenomenological model

Induction profiles were simulated from $E_{10}$ and $F_{90}$ metrics using functions that describe the induced fraction and the mean expression level of the induced subpopulations as a function of the glucose concentration. For the mean induced level, the following function was used:

$$\log_{10}(\text{Mean induced level}) = \frac{(\beta E_{10})^n}{[\text{glucose}]^n + (\beta E_{10})^n} * 2.5 - 3$$

where $\beta = \sqrt[n]{0.6/0.4}$ converts the glucose concentration where 10% of the maximal expression level is reached (i.e. the $E_{10}$) to the glucose concentration where 50% of the maximal expression level is reached. The function was scaled from $-3$ to $-0.5$ to match the range of the experimental data. To obtain realistic versions for the n constant, this function was fitted to the induced level curves of natural isolates, the mean fitted n value was extracted for every natural isolate, and the mean of these values was used for simulations (induced level curve: 1.15, induced fraction curve: 1.69, see *Figure 3—figure supplement 6*).

For the induced fraction, the following function was used:

$$\text{Induced fraction} = \frac{(\alpha F_{90})^n}{[\text{glucose}]^n + (\alpha F_{90})^n}$$

where $\alpha = \sqrt[n]{0.9/0.1}$ converts the glucose concentration where 90% of the cells are induced (i.e. the $F_{90}$) to the glucose concentration where 50% of the cells are induced. This function was fitted to the induced level curves of bimodal natural isolates, the mean fitted n value extracted for every natural isolate, and the mean of these values was used for simulations (*Figure 3—figure supplement 6*).

For nine concentrations of glucose and galactose, induced level and induced fraction values were extracted from these curves to generate simulated populations in these conditions. For a total population size of 20,000 cells, uninduced and induced subpopulations were generated according to the induced fraction value. The expression level values of cells were drawn from normal distributions with the mean expression level of the uninduced subpopulation at a constant level of $10^{-3}$ and the mean expression level of the induced subpopulation determined by the equation above. The standard deviations of the distributions were determined by fitting a quadratic equation to experimental standard deviations at different expression levels:

$$\text{Standard deviation} = -0.2 * (\log_{10}(\text{Induced level}) + 1.75)^2 + 0.4$$

To delineate possible unimodal and bimodal regimes, GAL induction was simulated using all possible combinations of 10 different values for $E_{10}$ and $F_{90}$ (1, 0.5, 0.25, 0.125, 0.0625, 0.0313, 0.156, 0.0078, 0.0039, 0.0020, 0.0010). The Hill constants n for the induced fraction and the induced level functions in these simulations were varied between the lowest and highest experimentally fitted n values (induced level curve: 0.84 and 1.50, induced fraction curve: 0.75 and 2.95, see *Figure 3—figure supplement 6*). The $E_{10}$ metric, the $F_{90}$ metrics, and the modality of the induction profile were

determined from these simulations as described above. In the $F_{90}$-$E_{10}$ space, unimodal and bimodal regimes were delineated by the bounding line with a slope of 1 that would capture all the unimodal or bimodal simulations respectively on one side of the line.

## Additional information

### Funding

| Funder | Grant reference number | Author |
|---|---|---|
| National Science Foundation | MCB-1349248 | Jue Wang<br>Michael Springer |
| National Institutes of Health | GM120122 | Julius Palme<br>Michael Springer |
| National Science Foundation | DGE1144152 | Jue Wang |

The funders had no role in study design, data collection and interpretation, or the decision to submit the work for publication.

### Author contributions

Julius Palme, Jue Wang, Conceptualization, Data curation, Formal analysis, Investigation, Visualization, Methodology, Writing - original draft, Writing - review and editing; Michael Springer, Conceptualization, Supervision, Funding acquisition, Investigation, Writing - original draft, Project administration, Writing - review and editing

### Author ORCIDs

Julius Palme ⓘD https://orcid.org/0000-0001-5897-1334
Michael Springer ⓘD https://orcid.org/0000-0002-3970-6380

### Decision letter and Author response

Decision letter https://doi.org/10.7554/eLife.69974.sa1
Author response https://doi.org/10.7554/eLife.69974.sa2

## Additional files

### Supplementary files

• Transparent reporting form

### Data availability

All data is deposited in a Dryad repository (https://doi.org/10.5061/dryad.69p8cz8z8).

The following dataset was generated:

| Author(s) | Year | Dataset title | Dataset URL | Database and Identifier |
|---|---|---|---|---|
| Springer M | 2021 | Data from: Variation in the modality of a yeast signaling pathway is mediated by a single regulator | https://doi.org/10.5061/dryad.69p8cz8z8 | Dryad Digital Repository, 10.5061/dryad.69p8cz8z8 |

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
