## [Decision Letter]

[Editors’ note: the authors submitted for reconsideration following the decision after peer review. What follows is the decision letter after the first round of review.]

Thank you for submitting your work entitled "Variation in the modality of a yeast signaling pathway is mediated by a single regulator" for consideration by *eLife*. Your article has been reviewed by 3 peer reviewers, one of whom is a member of our Board of Reviewing Editors, and the evaluation has been overseen by a Senior Editor. The reviewers have opted to remain anonymous.

Our decision has been reached after consultation between the reviewers. Based on these discussions, we regret to inform you that your work will not be considered further for publication in *eLife*.

Here is the consensus review:

This paper continues the lab's study of GAL1 induction in natural and laboratory strains of budding yeast. As previously shown, whether the induction is bimodal or unimodal depends upon the strain and the pre-induction carbon source. As the authors acknowledge in the Discussion section, we don't yet know the physiological significance of the phenomenon – i.e., is a unimodal response better under some environmental conditions and a bimodal response better under others – but still, working out how it arises is a nice test of our understanding of the system.

Here they present more data on the effects of strain background and pre-induction carbon source on the uni/bimodal nature of the response. They also present an ODE model of bistability and a much simpler phenomenological model that can account for many of the responses. This simpler model is based on Hill function (monostable) response curves with different thresholds for induced fraction and expression level of pathway output. And finally, they do gene swap experiments that argue that the uni vs. bimodal nature of a strain's response is largely determined by the sequence of the Gal3 gene.

This work could well be an interesting exercise in quantitative biology, but the reviewers found it so hard to understand what the authors were saying and showing that we are not really sure. For example, is the GAL system bistable (which is how the ODE model behaves) or not (as the phenomenological model assumes)? The reviewers were left confused about what the authors are concluding about the mechanism underpinning the phenomena they are describing.

We aim to publish work if revisions can be carried out in a couple of months. While required changes to the current work are too extensive to invite revision, we would likely be interested in a better-analyzed and better-presented take on this subject. It would be treated as a new submission, but we would try to recruit the same reviewers for evaluation.

Here are some specific aspects of the paper we had trouble with.

Data interpretation:

1. One of the core findings is that while all isolates are bimodal when galactose is titrated (in a background of raffinose), some are unimodal when glucose is titrated (in a background of an intermediate galactose (0.25%) + raffinose). One thing they did not mention, but can be seen comparing Figure S1 and Figure S2 is that some strains become more bimodal (in the sense that there is wide range of Glu/Gal mixtures in which two populations can clearly be detected). See for example the WashU strains or YJM981. (Does the ODE model also reproduce this?).

In general, this raises the question as to what is really going on. In my opinion, there might be a sort of misinterpretation of the data. Most strains seem to me to stay bimodal (comparing Figure S1 with S2), even if the statistical analysis they perform says otherwise. Let's take the case of the lab strain S288C as an example. The authors find that it is unimodal in Glu/Gal: however, what I see is that a group of uninduced cells persists as one moves from right to left (Figure S2), at the same time that the induced group starts to be induced in a graded fashion (again from right to left). So, in my view, that is a bimodal behavior: one mode corresponds to the group of uninduced yeast, and the other mode to the group of inducible cells, whose induction (or repression by glucose) is graded. The same behavior may be observed in other strains. My conclusion, comparing the galactose titration vs the glucose titration, is that the change in modality is (at least in most cases, maybe in all) apparent, not real; there is a change from a switch-like response to gradual induction (or repression) for those yeast that get induced, all in the context of a bimodal behavior. Thus, glucose is affecting only the inducible group.

In summary: the dose response to galactose (Figure S1) is switch-like (and bimodal), and the dose-response to glucose (Figure S2) (in a background of galactose) is graded (and bimodal).

If my interpretation is correct, the data in all the manuscript might need an overall change in interpretation.

Controls:

2. The Gal3 swap experiments are arguably the most interesting part of the paper (although, curiously, they are not mentioned in the abstract). And Gal3 was chosen for the swap for a good reason. However, it is quite possible that the other major regulators also affect strain behavior, and they could well be correlated with the allelic form of Gal3. As the authors know, previous work showed that simultaneous removal of the Gal3 and Gal1 positive feedbacks was required to truly eliminate bimodality. I wonder then what is the role of Gal1 and also Gal4 in strain to strain differences, since all these molecules have co-evolved in these strains. Thus, I think it would be important to show (a), considering that Gal1 serves a role very similar to Gal3, that Gal1 alleles are not important factors; (b) the result of a swap experiment using the Gal4 alleles, at least for a few interesting strains. Combining a joint swap of Gal3 and Gal4 and comparing with just Gal3 (already done with just Gal4). It would be important to see if the effect is reversed, or enhanced.

3. The authors need to present more than 'representative examples of at least two independent repeats'. Some assessment of experiment-to-experiment variability needs to be included.

The ODE model:

4. The ODE model needs to be written out. There is a parameter table, but without knowing what the rate equations are, the parameters are of little use. And as it is a reader can't really see what assumptions go into the model (e.g. Michaelis-Menten kinetics, which assume that the substrate is in huge excess over the enzyme?).

5. If I understand the ODE model correctly, it is a single-cell model; the authors are not trying to account for the cell-to-cell variability that makes the population level responses (sometimes) be bimodal. Why is this consideration included in the phenomenological model but not in the ODE model?

6. Finally, what is being measured is GAL1pr-YFP expression. What is being modeled in Figure 2 is various aspects of Gal4p and Gal3p. This is confusing.

The phenomenological model:

7. As mentioned above, this simpler model is based on Hill function (monostable) response curves (not bistable response curves, although I'm not sure how many readers will understand that the way this is written) with different thresholds for induced fraction and expression level of pathway output. And it accounts for much of the observed behavior. What does this mean? Is the point that the system is not bistable after all; or that the system may be bistable but you don't need bistability to account for the observed phenomena; or something else?

Clarity:

8. The authors need a more detailed cartoon than that shown in Figure 2A to give the uninitiated an idea of how the system works, and the scheme should include GAL1. The scheme also needs to be explained better.

9. If the authors are going to use the same figure panel more than once (e.g. Figure 7EF), the repetitions must be explicitly acknowledged.

10. Are the panels in Figure 6B flipped?

11. Throughout: Is it possible that 8 hours is too little to actually reach steady state after switching from pre-induced conditions? Could that explain the differences in strains? Maybe longer waiting needs to be tested.

12. Why are the GAL1pr-YFP fluorescence measurements normalized by dividing by SSC (a measure of cell texture) rather than FSC (a measure of cell size)?

13. Figure 2: Both Figure 2C and 2D are glucose titrations with constant galactose, so the labeling is confusing.

14. Line 171: '…determine whether a strain is bimodal' – bimodality is shown in many figures to depend on the pre-incubation conditions, not just the strain's identity. So what is meant by 'a strain is bimodal' – bimodal some of the time, all of the time, under some specific conditions compared across strains?

15. p. 9: The authors need to better explain why the fraction of active Gal3p should determine the fraction of cells in the induced state, whereas the amount of free Gal4p determines level of GAL1 induction in the induced cells. The logic is not apparent from Figure 2A. On p. 11 the authors do mention that they "previously showed that induced fraction and expression level are regulated by galactose/glucose ratio or the glucose concentration, respectively", but if "Pathway activation" is determined by Gal4p (Figure 2A) it is not clear how Gal3p and Gal4p could be determining different aspects of the response.

---

## [Author Response]

[Editors’ note: the authors resubmitted a revised version of the paper for consideration. What follows is the authors’ response to the first round of review.]

Here is the consensus review:This paper continues the lab's study of GAL1 induction in natural and laboratory strains of budding yeast. As previously shown, whether the induction is bimodal or unimodal depends upon the strain and the pre-induction carbon source. As the authors acknowledge in the Discussion section, we don't yet know the physiological significance of the phenomenon – i.e., is a unimodal response better under some environmental conditions and a bimodal response better under others – but still, working out how it arises is a nice test of our understanding of the system.Here they present more data on the effects of strain background and pre-induction carbon source on the uni/bimodal nature of the response. They also present an ODE model of bistability and a much simpler phenomenological model that can account for many of the responses. This simpler model is based on Hill function (monostable) response curves with different thresholds for induced fraction and expression level of pathway output. And finally, they do gene swap experiments that argue that the uni vs. bimodal nature of a strain's response is largely determined by the sequence of the Gal3 gene.This work could well be an interesting exercise in quantitative biology, but the reviewers found it so hard to understand what the authors were saying and showing that we are not really sure. For example, is the GAL system bistable (which is how the ODE model behaves) or not (as the phenomenological model assumes)? The reviewers were left confused about what the authors are concluding about the mechanism underpinning the phenomena they are describing.

We thank the reviewer for their comments. We have made substantial changes to the manuscript in order to improve clarity. We will address how bistability in the GAL pathway relates to the phenomenological model in our response to point 7.

Overall, the reviewers’ conclusion was that the manuscript could be ‘an interesting exercise in quantitative biology’, but the reviewers found it ‘hard to understand what the authors were saying and showing’. We believe that most of the confusion stems from our use in the original manuscript of two different types of models (mechanistic and phenomenological), and a lack of clarity in our explanation of how they relate to each other. Since we believe that the phenomenological model can capture the crucial features of variation in modality on its own, in the revised manuscript we have decided to remove the mechanistic model entirely. Instead, we significantly expanded and improved our explanation of the phenomenological model. We want to emphasize that the phenomenological model is on a solid mechanistic footing even in the absence of a formal ODE model, as we have described the mechanisms by which the features of the phenomenological model are regulated in another manuscript (see lines 87-95 in the manuscript and our response to point 15 for a description of these findings). We believe that these changes significantly improve the clarity of the modeling section while still providing a description of the underlying mechanisms.

In addition, the reviewers pointed out several additional control experiments such as additional allele swaps and extended incubation periods. We have included all the experiments in the revised manuscript and found that the results do not change the conclusions of the paper. Overall, we believe that the revised manuscript is a substantial improvement over the initial submission and hope that you find this revised version suitable for publication in *eLife*. Please see below for point-to-point answers to the concerns that were brought up by the reviewers.

Here are some specific aspects of the paper we had trouble with.Data interpretation:1. One of the core findings is that while all isolates are bimodal when galactose is titrated (in a background of raffinose), some are unimodal when glucose is titrated (in a background of an intermediate galactose (0.25%) + raffinose). One thing they did not mention, but can be seen comparing Figure S1 and Figure S2 is that some strains become more bimodal (in the sense that there is wide range of Glu/Gal mixtures in which two populations can clearly be detected). See for example the WashU strains or YJM981. (Does the ODE model also reproduce this?).

We agree that this is an interesting behavior. We have some hypotheses, but they involve experiments that are distinct from the ones performed here. As this observation is orthogonal to the findings we focus on in this manuscript, we hope to follow up on this observation in the future.

In general, this raises the question as to what is really going on. In my opinion, there might be a sort of misinterpretation of the data. Most strains seem to me to stay bimodal (comparing Figure S1 with S2), even if the statistical analysis they perform says otherwise. Let's take the case of the lab strain S288C as an example. The authors find that it is unimodal in Glu/Gal: however, what I see is that a group of uninduced cells persists as one moves from right to left (Figure S2), at the same time that the induced group starts to be induced in a graded fashion (again from right to left). So, in my view, that is a bimodal behavior: one mode corresponds to the group of uninduced yeast, and the other mode to the group of inducible cells, whose induction (or repression by glucose) is graded. The same behavior may be observed in other strains. My conclusion, comparing the galactose titration vs the glucose titration, is that the change in modality is (at least in most cases, maybe in all) apparent, not real; there is a change from a switch-like response to gradual induction (or repression) for those yeast that get induced, all in the context of a bimodal behavior. Thus, glucose is affecting only the inducible group.In summary: the dose response to galactose (Figure S1) is switch-like (and bimodal), and the dose-response to glucose (Figure S2) (in a background of galactose) is graded (and bimodal).If my interpretation is correct, the data in all the manuscript might need an overall change in interpretation.

We disagree that there are no unimodal strains. We would argue that the population distribution of responses of the following strains in Figure 1—figure supplement 1 is clearly unimodal regardless of criteria: Bb32, BC187, CLIB215, FL100, I14, NC-02, T7, YJM653, YPS163, YPS606, and YPS1009. There are additional strains that are highly likely to be unimodal but there is some ambiguity caused by dead/dying cells (cells that don’t induce regardless of carbon ratio): IL-01 and YPS128.

The crux of the comment is about when to call a population distribution of response unimodal or bimodal. From a mathematical standpoint, the distinction between monostable and bistable is unambiguous. But here we are discussing unimodal and bimodal curves in the presence of experimental noise. While these results can hint at underlying stability they do not strictly correlate and hence there is no strict criteria by which to call something unimodal or bimodal. For example, if one in a billion cells behaved differently, the system is technically bimodal but the physiological relevance is unclear, the cell might not be detected, and the rare cell could even be an artifact. Because of this we are forced to pick metrics (cut-offs) which capture and stratify the range of behaviors. The reviewer’s comment highlighted one of the most ambiguous induction behaviors, S288C after pre-induction growth in raffinose, which we agree is on the borderline of a unimodal versus bimodal call. For the analysis in this manuscript, we decided that an appreciable fraction of the population (15%) must be in the smaller subpopulation (in any condition) for a population distribution to be called bimodal. To address the concerns presented here, we tested a range of cut-offs and our results were largely unaffected (Figure 3—figure supplement 8). S288C is in the minority of strains that can be affected by reasonable changes in metrics; switching the modality call for this strain does not significantly affect the results of this manuscript.

To improve clarity, we have included histograms of intermediate GAL induction in unimodal vs bimodal induction profiles in Figure 1B. Further, we included histograms of all analyzed conditions for the glucose titrations from Figure 1 in a new panel in Figure 1—figure supplement 4. These plots clearly show that some induction profiles never display bimodal GAL induction. We hope that the new plots better illustrate the difference between bimodal and unimodal behavior.

Controls:2. The Gal3 swap experiments are arguably the most interesting part of the paper (although, curiously, they are not mentioned in the abstract). And Gal3 was chosen for the swap for a good reason. However, it is quite possible that the other major regulators also affect strain behavior, and they could well be correlated with the allelic form of Gal3. As the authors know, previous work showed that simultaneous removal of the Gal3 and Gal1 positive feedbacks was required to truly eliminate bimodality. I wonder then what is the role of Gal1 and also Gal4 in strain to strain differences, since all these molecules have co-evolved in these strains. Thus, I think it would be important to show (a), considering that Gal1 serves a role very similar to Gal3, that Gal1 alleles are not important factors; (b) the result of a swap experiment using the Gal4 alleles, at least for a few interesting strains. Combining a joint swap of Gal3 and Gal4 and comparing with just Gal3 (already done with just Gal4). It would be important to see if the effect is reversed, or enhanced.

We thank the reviewers for their comment. We have changed the abstract to highlight the *GAL3* allele swap experiments and performed the additional allele swap experiments. In Figure 6—figure supplement 2, we show that in the S288C background *GAL1* or *GAL4* allele swaps do not change modality (although some effect on the induced mean can be seen). In Figure 6—figure supplement 3, we show that in the S288C background joint swaps of either *GAL1* or *GAL4* with the respective *GAL3* allele do not affect modality beyond what was observed when *GAL3* was swapped alone.

In the main text we added the following sentence to highlight these results:

“Swapping the alleles of GAL1, the second galactose sensor, or GAL4, the transcription factor that activates GAL gene expression, does not affect modality (Figure 6—figure supplement 6). Similarly, swaps of GAL1 or GAL4 alleles in GAL3 allele swap strains have no additional effect on modality (Figure 6—figure supplement 4), suggesting that the role of GAL3 in setting modality can evolve independently of other GAL genes.”

3. The authors need to present more than 'representative examples of at least two independent repeats'. Some assessment of experiment-to-experiment variability needs to be included.

We agree we should have included more examples to demonstrate the reproducibility of the results. To this end we have included assessments of variability for the glucose gradients of the natural isolates and for the pre-induction condition and GAL3 allele swap experiments. In Figures 1—figure supplement 5, 4—figure supplement 3 and 6—figure supplement 4, we plot the induced fraction and induced mean of replicate measurements as a function of the glucose concentration. In Figure 3—figure supplement 7, we include the standard deviation of at least two E_10_ and F_90_ measurements in the E_10_-F_90_ phase space plotted in Figure 3E. In Figures 4—figure supplement 4 and 6—figure supplement 5, we plot the reproducibility of two independent E_10_ and F_90_ measurements for changing pre-induction conditions and *GAL3* allele swaps.

The ODE model:4. The ODE model needs to be written out. There is a parameter table, but without knowing what the rate equations are, the parameters are of little use. And as it is a reader can't really see what assumptions go into the model (e.g. Michaelis-Menten kinetics, which assume that the substrate is in huge excess over the enzyme?).5. If I understand the ODE model correctly, it is a single-cell model; the authors are not trying to account for the cell-to-cell variability that makes the population level responses (sometimes) be bimodal. Why is this consideration included in the phenomenological model but not in the ODE model?6. Finally, what is being measured is GAL1pr-YFP expression. What is being modeled in Figure 2 is various aspects of Gal4p and Gal3p. This is confusing.

As explained above, we have removed the ODE model from the paper.

The phenomenological model:7. As mentioned above, this simpler model is based on Hill function (monostable) response curves (not bistable response curves, although I'm not sure how many readers will understand that the way this is written) with different thresholds for induced fraction and expression level of pathway output. And it accounts for much of the observed behavior. What does this mean? Is the point that the system is not bistable after all; or that the system may be bistable but you don't need bistability to account for the observed phenomena; or something else?

The Hill function for the induced fraction describes the frequency distribution between the two bimodal states (uninduced and induced states). This function assumes the coexistence of two stable states, a starting assumption that is supported by the previous work showing bistability of the GAL pathway response (reference 10). We have completely rewritten the section describing the phenomenological model to reflect this and hope that this point is communicated more clearly now.

Clarity:8. The authors need a more detailed cartoon than that shown in Figure 2A to give the uninitiated an idea of how the system works, and the scheme should include GAL1. The scheme also needs to be explained better.

We have moved the cartoon the Figure describing the phenomenological model (Figure 2A). We have included regulation of GAL promoters in the cartoon.

9. If the authors are going to use the same figure panel more than once (e.g. Figure 7EF), the repetitions must be explicitly acknowledged.

We have removed this repetition (see new Figure 6).

10. Are the panels in Figure 6B flipped?

We are not sure which panels the reviewer is referring to, but we verified the order the panels in the new Figure 5.

11. Throughout: Is it possible that 8 hours is too little to actually reach steady state after switching from pre-induced conditions? Could that explain the differences in strains? Maybe longer waiting needs to be tested.

This is a very reasonable concern. For the induced fraction of bimodal induction profiles, we have previously shown that YFP expression from the GAL1 promoter reaches steady-state levels after 8 hours (see reference 23). Here, we measured YFP expression levels at both 8 hours and 24 hours modality for 14 natural isolates with unimodal induction profiles after pre-induction growth in raffinose and verified that there are no changes in modality (Figure S2).

12. Why are the GAL1pr-YFP fluorescence measurements normalized by dividing by SSC (a measure of cell texture) rather than FSC (a measure of cell size)?

This perception comes largely from people sorting mixtures of immunological cells. While FSC is a good measure of cell size when comparing different cell types, we (reference 23) and others (10.1038/nature04785) have found that within a cell type, SSC is a better measure for cell size than FSC (based on the correlation between SSC or FSC and the fluorescence of a constitutively expressed fluorescent protein).

13. Figure 2: Both Figure 2C and 2D are glucose titrations with constant galactose, so the labeling is confusing.

We have removed the old Figure 2 from the paper.

14. Line 171: '…determine whether a strain is bimodal' – bimodality is shown in many figures to depend on the pre-incubation conditions, not just the strain's identity. So what is meant by 'a strain is bimodal' – bimodal some of the time, all of the time, under some specific conditions compared across strains?

We have changed the wording from ‘whether a strain is bimodal’ to ‘whether the induction behavior is bimodal’ in order to reflect that modality depends on the pre-induction conditions.

15. p. 9: The authors need to better explain why the fraction of active Gal3p should determine the fraction of cells in the induced state, whereas the amount of free Gal4p determines level of GAL1 induction in the induced cells. The logic is not apparent from Figure 2A. On p. 11 the authors do mention that they "previously showed that induced fraction and expression level are regulated by galactose/glucose ratio or the glucose concentration, respectively", but if "Pathway activation" is determined by Gal4p (Figure 2A) it is not clear how Gal3p and Gal4p could be determining different aspects of the response.

In a manuscript that was just published (updated reference 22 ), we show that the different aspects of the response are determined by either glucose or the galactose/glucose ratio. Even though *GAL1* expression is solely determined by the amount of free Gal4p, glucose and galactose can have independent effects: The galactose/glucose ratio regulates Gal3p activity and thereby, whether cells are the induced state or not (which we analyze here as the ‘induced fraction’). The glucose concentration regulates *GAL4* expression and thereby, what the expression level is if cells are in the induced state. While we have removed the ODE model from the paper, we believe that this concept provides an interesting mechanistic explanation for why in the phenomenological model the induced fraction and induced mean regulation can be varied independently. We have completely rewritten the section that describes these results and hope that the new section provides a much clearer explanation.